# Operational complexities in international clinical trials: a systematic review of challenges and proposed solutions

Leher Gumber [1,2] Opeyemi Agbeleye,[3] Alex Inskip,[3] Ross Fairbairn,[3] Madeleine Still,[3] Luke Ouma,[4] Jingky Lozano-Kuehne,[4] Michelle Bardgett,[5] John D Isaacs,[1,6] James MS Wason,[4] Dawn Craig,[3] Arthur G Pratt [1,6]

DC and AGP contributed equally.

For numbered affiliations see end of article.

**Correspondence to**
Dr Arthur G Pratt;
arthur.pratt@newcastle.ac.uk

## ABSTRACT

**Objective** International trials can be challenging to operationalise due to incompatibilities between country-specific policies and infrastructures. The aim of this systematic review was to identify the operational complexities of conducting international trials and identify potential solutions for overcoming them.

**Design** Systematic review.

**Data sources** Medline, Embase and Health Management Information Consortium were searched from 2006 to 30 January 2023.

**Eligibility criteria** All studies reporting operational challenges (eg, site selection, trial management, intervention management, data management) of conducting international trials were included.

**Data extraction and synthesis** Search results were independently screened by at least two reviewers and data were extracted into a proforma.

**Results** 38 studies (35 RCTs, 2 reports and 1 qualitative study) fulfilled the inclusion criteria. The median sample size was 1202 (IQR 332–4056) and median number of sites was 40 (IQR 13–78). 88.6% of studies had an academic sponsor and 80% were funded through government sources. Operational complexities were particularly reported during trial set-up due to lack of harmonisation in regulatory approvals and in relation to sponsorship structure, with associated budgetary impacts. Additional challenges included site selection, staff training, lengthy contract negotiations, site monitoring, communication, trial oversight, recruitment, data management, drug procurement and distribution, pharmacy involvement and biospecimen processing and transport.

**Conclusions** International collaborative trials are valuable in cases where recruitment may be difficult, diversifying participation and applicability. However, multiple operational and regulatory challenges are encountered when implementing a trial in multiple countries. Careful planning and communication between trials units and investigators, with an emphasis on establishing adequately resourced cross-border sponsorship structures and regulatory approvals, may help to overcome these barriers and realise the benefits of the approach.

**Open science framework registration number** osf-registrations-yvtjb-v1.

## STRENGTHS AND LIMITATIONS OF THIS STUDY

⇒ A robust search and screening strategy was used to maximise the identification of relevant studies.
⇒ All abstracts were screened by at least two reviewers.
⇒ The review focused on international trials so the findings may not be generalisable to multi-centre studies conducted within the same country.
⇒ The possibility that some specific challenges encountered might be omitted cannot be entirely excluded nor can an element of subjectivity when summarising potential solutions to those identified.

## INTRODUCTION

The development and deployment of international clinical trials that enrol participants from more than one nation or jurisdiction continue to increase.[1 2] Motivated by advancements in technology, globalisation and insufficient accrual rates using traditional approaches,[3] recent examples increasingly adopt master protocols that allow treatment arms to be added and dropped adaptively over time—so-called 'platform' trials—and other innovative designs.[4] International trials offer numerous advantages over single-nation approaches by increasing access to potentially eligible participants, enabling faster recruitment and/or larger sample sizes while encouraging representation of diverse ethnic, biological and socio-cultural groups.[3 5–7] They promote best practice globally and expand horizons for treatment availability in countries that may not otherwise have access to particular interventions.[7] The reduced operational cost of running a trial in developing compared with developed countries may be an additional consideration.[8] They furthermore foster collaboration and

development of closer relationships among academics globally.[3 5 7 9 10]

The conduct of international trials nonetheless presents unique challenges. Their success requires adherence to local and international laws, regulations and ethical requirements, availability of Good Clinical Practice (GCP)-trained researchers, adequate infrastructure at clinical sites, and close monitoring and oversight across multiple jurisdictions.[7 11 12] Incompatibilities between country-specific policies and challenges in the trial setup were previously identified as factors that extend study timelines and inflate costs.[13] Other contributing factors including site selection, insurance, logistics, regulatory requirements and sponsorship have been highlighted in a narrative review.[14] Platform trials present further complexity, potentially involving adaptations that incur ethical and/or regulatory review, sample size re-estimations, site capacity and data management challenges.[15 16] In recognition of these issues, we conducted a systematic review that aimed to identify (1) the operational challenges of conducting international clinical trials and (2) potential practical solutions for overcoming them. Our overarching objective was to deliver a reference of practical value to prospective trialists planning to set up international trials in the modern era, with the potential to inform best practice guidelines.

## METHODS

This study was reported in accordance with the Preferred Reporting Items for Systematic Reviews and Meta-analyses (PRISMA) guidelines.[17] The study was registered with the Open Science Framework (Identifier: osf-registrations-yvtjb-v1)[18] and funded by National Institute of Health and Care Research (NIHR153955).

### Search strategy and study selection

A search strategy was developed in collaboration with an information specialist (AI) using a combination of key words and MeSH terms. Medline and Embase were the primary bibliographic database sources, but a variation of the search was also run on the Health Management Information Consortium (HMIC) database. The search covered four broad concepts: (i) international trials, (ii) adaptive trials (in recognition that such trials frequently adopt international enrolment strategies not explicit in titles/abstracts), (iii) specific challenges to conducting trials and (iv) a focus on study design methods. These concepts were used in several different combinations to achieve a practicable quantity of relevant results while mitigating the risk of omitting relevant material. Studies conducted prior to 2006 were considered less likely to reflect current legislation and hence a limit was applied to exclude them in the initial search strategy, as were systematic reviews and studies not available in English. A full description of the search strategy is provided in the online supplemental file.

After de-duplication of records, titles and abstracts were rigorously double screened by at least two reviewers. The following studies were excluded: protocols, abstracts, multicentre trials in a single country, studies published before 2006, studies not in English, systematic reviews and studies which did not report operational challenges. Operational challenges broadly encompassed issues related to approvals, opening sites, recruitment and data management and are further detailed in the online supplemental file.

For studies deemed eligible, or where it was deemed impossible to decide eligibility from the abstract, the full text was retrieved. Full-text articles were screened for inclusion by one reviewer and 20% of the full-text articles underwent screening by a second reviewer. In case of discrepancy, the decision was taken by consensus of all the reviewers. Rayyan (Qatar Computing Research Institute), a systematic review web-based application, was used for screening, with record management facilitated throughout using the Zotero reference management tool.

### Data extraction

Data were extracted by one reviewer using a pre-designed proforma (online supplemental table S1) and extractions were checked by a second reviewer. Aside from general trial characteristics, data extraction was under the broad headings: sponsorship, funding, regulatory considerations, trial management, intervention, biospecimens and agreements. Due to the nature of the review, a formal quality assessment was not performed.

### Definition of variables and data analysis

The sponsor was determined as either academic (universities, hospitals or the government) or industry (pharmaceutical or device companies). The funder was defined as academic (university or hospital), government, industry or charity; where studies were co-funded by more than one source, the predominant funder was reported. Location of studies was categorised by continent into UK/Europe (EU), Asia, Africa, North America, South America or Oceania. Primary outcome measures were considered as efficacy, effectiveness, prevention or safety. Quantitative data were described using numbers, percentages, median and IQR. Qualitative data were summarised into overarching themes which described the main challenges of conducting international trials. In relation to specific challenges identified during data extractions, solutions given by the authors were also recorded where available in each case and synthesised for reporting purposes.

### Patient and public involvement

No patients were involved.

## RESULTS

The search identified 5215 records, of which 1588 were duplicates and removed. After title and abstract screening of the remaining 3627 articles, a further 3374 were

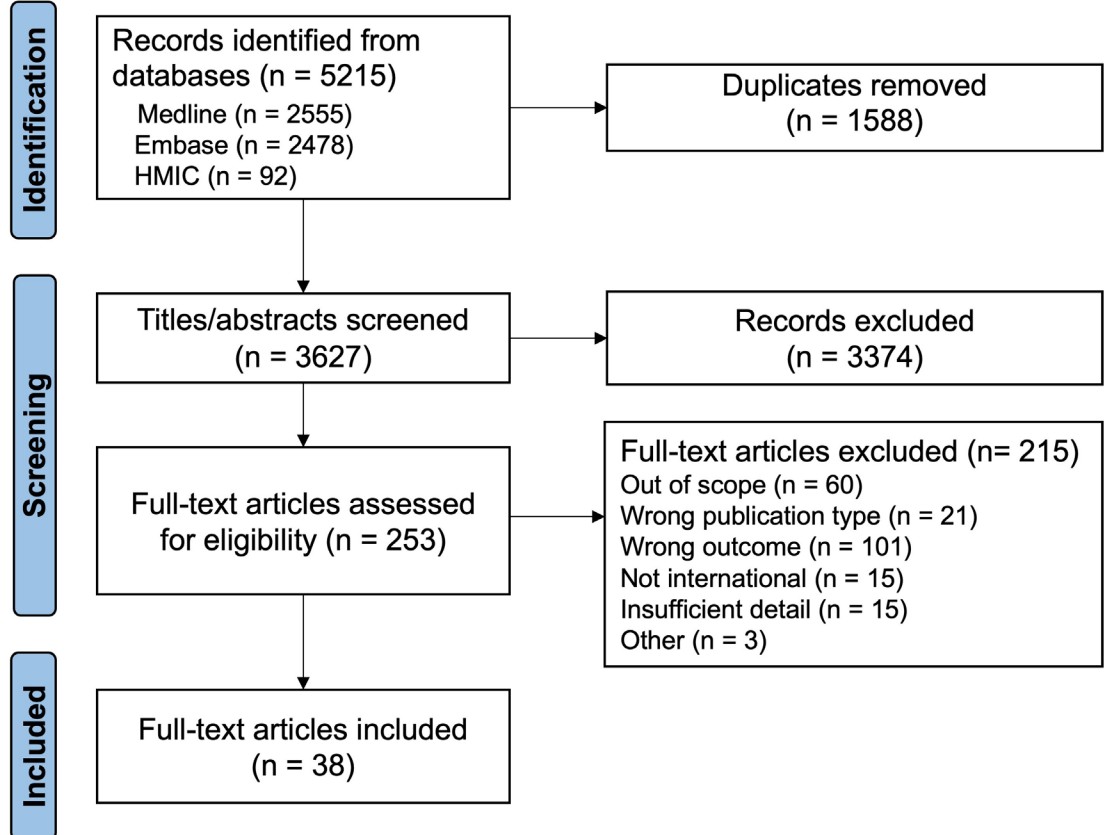

**Figure 1** PRISMA flow diagram.

excluded. Full-text screening was conducted on 253 articles from which 38 were included (figure 1). Characteristics of included studies are summarised in table 1 and further details are available in online supplemental table S2.

### Study characteristics

Of the 38 included studies, 35 (92.1%) were RCTs, 2 (5.3%) were reports and 1 (2.6%) was a qualitative study. Of the 35 RCTs, 24 (68.6%) had a parallel design, 7 (20.0%) were adaptive, 3 (8.6%) were factorial and 1 (2.9%) was cluster. Over half of the RCTs were open-label. Most RCTs evaluated a drug intervention (74.3%) and measured efficacy (40.0%) as their primary outcome. The median number of sites and sample size among closed trials was 40 (IQR 13–78) and 1202 (IQR 332–4056) respectively. The majority of studies (n=31, 88.6%) were sponsored by an academic institution and the most common funding source was government (80.0%). The continents most commonly represented by sites included in studies were North America (n=27, 77.1%) and UK/EU (n=25, 71.4%). Design characteristics of the included studies are summarised in table 2. Operational complexities of conducting international trials were reported broadly at six stages of the trial process including study set-up, site set-up, trial management, data management,

intervention management and adaptive specific features (figure 2; online supplemental table S3). We structure our findings accordingly herein, summarising recurring themes and potential approaches arising to address them in table 3.

### Study set-up

#### Sponsorship, insurance and need for EU legal representative

13 studies[11] [19–28] described[29] [30] variations[31] [32] in[33] sponsorship[34–62] and insurance requirements between countries which created delays.[11] [19–30] This was particularly notable for UK/EU sites. The EU Clinical Trials Directive which governs the conduct of clinical trials in EU requires the presence of an EU legal representative for trials sponsored by a non-EU institution.[31] After identification of an EU legal representative, repeated negotiations were necessary to clarify roles and responsibilities between the sponsor and legal representative resulting in further delays.[19–21] [32] Country-specific variations in insurance and indemnity requirements within the EU were a further hurdle. For example, two studies noted that the minimum compensation in Germany was 500 000 euros and the insurance provided by the sponsor did not meet these limits. Additional insurance coverage could not be provided locally and, consequently, those sites were

**Table 1** Summary of included studies

| First author (year) | Trial name | Trial design | Population | Number of sites | Site locations | Main coordinating centre(s) | Total participants | Intervention |
|---|---|---|---|---|---|---|---|---|
| Aban[26] (2008) | MGTX | Parallel | Myasthenia gravis | 79 | Global | USA | 126 | Other |
| Aitken[33] (2008) | PROMOTION | Parallel | Coronary artery disease | 5 | North America, Oceania | USA | 3522 | Behavioural |
| Angus[34] (2020) | REMAP-CAP | Adaptive-platform | Severe pneumonia and COVID-19 | | UK/EU, North America, Oceania, Asia | Australia, Thailand | | Drug |
| Aryal[39] (2021) | REMAP-CAP | Adaptive-platform | Severe pneumonia and COVID-19 | | UK/EU, North America, Oceania, Asia | RCC in Australia and Thailand | | Drug |
| Antic[43] (2015) | SAVE | Parallel | Obstructive sleep apnoea | 89 | Oceania, North America, South America, Asia, UK/EU | RCC in Australia, Brazil, China, India and Spain | 2717 | Device |
| Babiker[19] (2013) | START | Parallel | HIV | 237 | North America, South America, UK/EU, Oceania, Africa | RCC in Denmark, UK, Australia, USA | 4000 | Drug |
| Berthon-Jones[27] (2015) | ALTAIR | Parallel | HIV | 36 | Asia, Oceania, UK/EU, North America, South America | USA | 322 | Drug |
| Bryant[50] (2021) | TBTC Study 31 | Parallel | Tuberculosis | 34 | North America, South America, Asia, Africa | USA | 2516 | Drug |
| Carli[44] (2013) | SEYLE | Cluster | Suicide | 11 | UK/EU | Sweden | 11110 | Behavioural |
| Clasen[54] (2020) | HAPIN | Parallel | Low birth weight | | Asia, North America, South America, Africa | USA | | Device |
| Coomarasamy[51] (2016) | PROMISE | Parallel | Recurrent miscarriage | 45 | UK/EU | UK | 836 | Drug |
| Crow[20] (2018) | FOR-DMD | Parallel | Duchenne muscular dystrophy | 40 | North America, UK/EU | USA | 196 | Drug |
| del Álamo[21] (2022) | | | | | | | | |
| Denholm[52] (2022) | ASCOT ADAPT | Adaptive-platform | COVID-19 | | Oceania, Asia | | | Drug |
| Dutton[45] (2009) | CONTROL | Parallel | Trauma | 75 | North America, South America, UK/EU, Asia, Africa | USA | 576 | Drug |
| Eikelboom[40] (2022) | ACT | Factorial | COVID-19 | 62 | North America, South America, Africa, Asia | USA | 6528 | Drug |
| Fogelholm[46] (2017) | PREVIEW | Factorial | Pre-diabetes | 8 | UK/EU, Oceania | | 2326 | Behavioural |
| Franciscus[53] (2014) | TRIGR | Parallel | Type 1 diabetes | 77 | North America, Oceania, UK/EU | USA | 5156 | Other |
| Fulda[49] (2023) | REPRIEVE | Parallel | HIV | | North America, South America, Africa, Asia, UK/EU | USA | | Drug |
| Goossens[28] (2022) | REMAP-CAP | Adaptive-platform | Severe pneumonia and COVID-19 | | UK/EU, North America, Oceania, Asia | Australia, Thailand | | Drug |

Continued

**Table 1** Continued

| First author (year) | Trial name | Trial design | Population | Number of sites | Site locations | Main coordinating centre(s) | Total participants | Intervention |
|---|---|---|---|---|---|---|---|---|
| Grarup[32] (2015) | START | Parallel | HIV | 237 | North America, South America, UK/EU, Oceania, Africa | RCC in Denmark, UK, Australia, USA | 4000 | Drug |
| Hata[41] (2021) | PATHWAY | Parallel | Breast cancer | 23 | Asia | | 185 | Drug |
| Herrick[35] (2012) | FDTT | Parallel | Functional dyspepsia | 8 | North America | | 292 | Drug |
| Jeon[47] (2016) | CLEAR III | Parallel | Intracerebral haemorrhage | 73 | North America, South America, UK/EU, Asia | USA | 500 | Drug |
| Kenyon[22] (2011) | STICH II | Parallel | Intracerebral haemorrhage | 126 | North America, Oceania, UK/EU, Asia, Africa | UK | 601 | Other |
| Kesho Bora Study Group[36] (2011) | Kesho-Bora | Parallel | HIV | 5 | Africa | Switzerland | 824 | Drug |
| Kolitsopoulos[48] (2013) | ZODIAC | Parallel | Schizophrenia | 226 | North America, South America, UK/EU, Asia | | 18240 | Drug |
| Larson[23] (2016) | INSIGHT trials | | | | | RCC in UK, Denmark, USA and Australia | | |
| Lingor[24] (2021) | ROCK-ALS / ROCK-ALS-US | Parallel | Amyotrophic lateral sclerosis | | North America, UK/EU | | | Drug |
| Minisman[7] (2012) | MGTX | Parallel | Myasthenia gravis | 79 | Global | USA | 126 | Other |
| Murray[25] (2022) | TICO | Adaptive-platform | COVID-19 | 51 | North America, UK/EU, Asia, Africa | USA with 8 RCC | | Drug |
| Neaton[11] (2010) | INSIGHT trials | | | | | RCC in UK, Denmark, USA and Australia | | |
| Ravinetto[37] (2013) | 4ABC | Parallel | Malaria | 12 | Africa | Belgium | 4112 | Drug |
| Reams[42] (2018) | DOVE | Parallel | Sickle cell disease | 51 | North America, South America, UK/EU, Asia, Africa | | 341 | Drug |
| Seal[29] (2006) | | Factorial | Endophthalmitis | 24 | UK/EU, Asia | UK | 35000 | Drug |
| Spencer[55] (2012) | AWARD-5 | Adaptive | Type 2 diabetes | 111 | North America, UK/EU, Asia | | 1202 | Drug |
| Sydes[30] (2012) | STAMPEDE | Adaptive-platform | Prostate cancer | | UK/EU | UK | | Drug |
| Zimmer[38] (2010) | BAMSG 3-01 | Parallel | Cryptococcal meningitis | 13 | North America, Asia | USA | 143 | Drug |

RCC, Regional Coordinating Centre; RCT, randomised controlled trial.

**Table 2** Design characteristics of included trials (n=35)

| Features | N (%) |
|---|---|
| Design | |
| Parallel | 24 (68.6) |
| Adaptive | 7 (20.0) |
| Cluster | 1 (2.9) |
| Factorial | 3 (8.6) |
| Masking | |
| Open label | 19 (54.3) |
| Blinded | 16 (45.7) |
| Number of sites among closed trials | |
| median (IQR) | 40 (13–78) |
| Continents of site locations | |
| Africa | 14 (40.0) |
| Asia | 22 (62.9) |
| North America | 27 (77.1) |
| Oceania | 14 (40.0) |
| South America | 14 (40.0) |
| UK/Europe | 25 (71.4) |
| Trial status | |
| Open | 9 (25.7) |
| Closed | 25 (71.4) |
| Terminated | 1 (2.9) |
| Sample size among closed trials | |
| median (IQR) | 1202 (332–4056) |
| Intervention | |
| Drug | 26 (74.3) |
| Device | 2 (5.7) |
| Behavioural | 3 (8.6) |
| Other | 4 (11.4) |
| Primary outcome | |
| Efficacy | 14 (40.0) |
| Effectiveness | 12 (34.3) |
| Safety | 2 (5.7) |
| Prevention | 7 (20.0) |
| Sponsor | |
| Academic | 31 (88.6) |
| Industry | 3 (8.6) |
| Unknown | 1 (2.9) |
| Main funder | |
| Academic | 1 (2.9) |
| Government | 28 (80.0) |
| Industry | 5 (14.3) |
| Charity | 1 (2.9) |

either dropped or authors had to run two parallel trials in different continents.[23 24]

## Funding

14 studies described funding as a challenge.[19–24 26 28 33–38] Interestingly, all of these were funded through government sources. Limitations in funding meant that some investigators had to seek additional sources of support and in cases where this was unsuccessful, there was a delay in trial start-up.[20 21 36 37] Additionally, trials funded by the USA had a requirement for non-US sites to obtain departmental clearance and approval from their respective countries before funds could be transferred. A lack of familiarity with this process prevented timely transfer of funds and thus delayed recruitment.[26] Predicting accurate budget projections and variations in currency exchange rates were additional challenges for trials spanning several years.[33] Prolonged negotiations between the funder, site and sponsor regarding site set-up costs led to further delays.[32]

## Lack of harmonisation in ethics and regulatory approvals

A lack of harmonisation in international legal and ethical systems was a reoccurring theme.[11 19 21 22 25–29 33 34 37–42] The time from initiation of regulatory procedures to the start of the trial ranged from 3 to 18 months. Hurdles included lack of a centralised system and therefore the requirement for single-centre approvals,[7 19 22 24 26 33 40] country-specific differences in requirements,[7 22 24 26 37 41 42] infrequent ethics committee meetings,[22 33] ethics review fee,[22 37] translation of essential documents,[41] protocol amendments[36 37] and lack of familiarity among ethics committees with trial design or conduct.[22 40] Additionally, multinational trials funded in total or in part by US National Institute of Health required Federalwide Assurance (FWA) approvals and annual reviews alongside adherence to country-specific regulations. Although this process was familiar to US sites, non-US sites found this challenging.[7 11 23 26]

## Site set-up
### Training

In general, staff training was not considered a challenge.[20 33 36 39 43–48] Studies used a combination of in-person and online workshops which covered a broad range of topics including study protocol, Good Clinical Practice, data management, intervention delivery and safety reporting.[20 26 34 46 47] One trial adopted a system of 'train-the-trainer,' whereby site leads would receive centralised training then train all personnel at their local sites.[44]

### Contracts

Contracts were formed at many levels including site agreements, funding agreements, sponsorship agreements, agreements with pharmaceutical companies and data sharing. Several studies described contract negotiations as a time consuming and lengthy process.[19–21 23 24 28 32] Contributing factors to this were translation of contracts,[26] conflict with country-specific legal terminology and interpretation,[20 21] clauses of indemnification[19 20] and administrative bureaucracy in legal departments.[32] Data sharing across borders requires additional levels of consent and protection. Trials with EU sites described how all participating sites were required to comply with the EU General

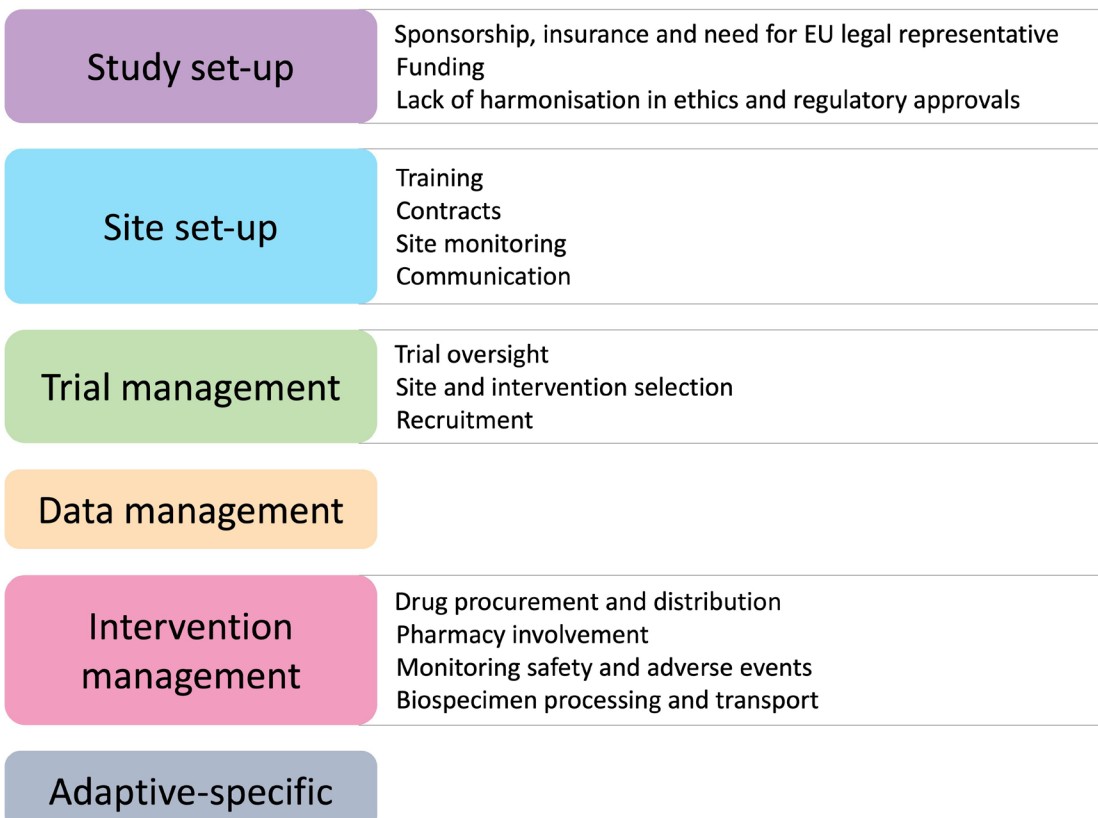

**Figure 2** Operational complexities of international trials.

Data Protection Regulations (GDPR).[21 24] This required additional administrative steps.[24] To overcome these challenges, INSIGHT (International Network for Strategic Initiatives in Global HIV Trials) established coordinating centres in several countries that were geographically closer to sites, familiar with local regulatory requirements and fluent in local languages which improved the contract negotiation process.[23]

### Site monitoring
Monitoring and auditing of sites was conducted periodically and included checking for compliance with the protocol and standard operating procedures (SOPs), data consistency and missing data. Most studies adopted a combination of remote and in-person quality assurance visits and created summary reports that were circulated to sites.[34 39 49–51] Underperforming sites were asked to submit an action plan and provide additional staff training.[49 51] A clear escalation procedure was established for ensuring timely improvement of site performances.[49] Studies reported the importance of planning adequate human and financial resources for these tasks.[29 37]

### Communication
Regular communication with sites and investigators was critical.[20 33–35 43 49 52] Multiple communication channels were used to ensure consistency across sites, overcome time zone and cross-cultural differences. These included emails for study protocols, newsletters, progress reports and site score cards, and regular verbal communication in the form of monthly site teleconferences, outreach calls with leadership team and biannual or annual face-to-face meetings.[7 20 33 35 43 49 52] Communications were usually managed by a contract research organisation, clinical trials unit or the site selection committee.[47 49] One study described in detail the guidelines they created to streamline the management of site queries and thereby limit unnecessary or excessive email traffic.[33]

### Trial management
#### Trial oversight
A number of management 'tiers' could be identified, according to their scope of responsibility for trial oversight. Most studies described a Central Coordinating Centre (also referred to as a 'trial coordinating centre', 'main operational centre' or 'clinical or data coordinating centre'; CCC) with over-arching responsibility for overseeing trial setup and conduct in accordance with the protocol, SOPs and regulatory approvals.[11 23 25 26 37 38 44 47 49–52] The CCC worked with at least one national coordinating centre (also referred to as a 'country coordinating centre' or 'national project manager') that supported regulatory approvals, set-up of sites and delivery of the trial in each country.[19 25 32 43] In trials spanning multiple continents, an additional level of management with an international coordinating centre worked closely with the CCC and was responsible for oversight of several

**Table 3** Challenges and proposed solutions

| | Challenges | Proposed solutions |
|---|---|---|
| Study set-up | Identification of appropriate sponsor and agreements | ▶ Clarify roles and responsibilities between sponsor, co-sponsor and legal representative in advance<br>▶ Ascertain insurance policy requirements prior to country and site selection<br>▶ Identify a lead site for each country/continent that is familiar with local regulations to act as a coordinating centre for all local sites |
| | Budget considerations | ▶ Establish site set-up costs early on (including start-up fees, pharmacy dispensing, lab, ethics, administration, archiving and close out)<br>▶ Consideration of changes in currency exchange rates<br>▶ Accurate budget predictions for trials spanning several years<br>▶ Translation costs |
| | Regulatory and ethical approvals | ▶ Ensure all sites are familiar with the process required for regulatory approvals |
| Site set-up | Contracts with collaborating sites | ▶ Begin contract negotiations early<br>▶ Establish coordinating sites or centres within each country or region to manage negotiations<br>▶ Allocate time and costs for translation<br>▶ Coordination with legal representatives to clarify country-specific terminology and interpretation of legal responsibilities and GDPR |
| | Site monitoring | ▶ Plan adequate financial and human resources for in-person visits<br>▶ Clear escalation procedures and remedial actions for sites which are underperforming |
| | Channels of communication | ▶ Regular communication with sites and investigators using multiple channels including emails, monthly teleconferences, outreach calls and annual face-to-face meetings<br>▶ Develop a hierarchical process for managing site queries to avoid unnecessary emails |
| | Translation | ▶ Plan adequate financial resources and time for translation of essential documents, contracts and drug packaging |
| Trial management | Site and intervention selection | ▶ Conduct site feasibility assessments during trial planning phase<br>▶ Use a systematic process to guide intervention selection and removal particularly for adaptive trials |
| | Recruitment and retention | ▶ Frequent, often weekly communication with study participants using different channels (eg, letters, phone calls, emails) |
| | | ▶ Reimbursement of transport costs<br>▶ Keeping study visits to a minimum |
| Data management | Technological challenges including internet bandwidth, institutional firewalls and device availability | ▶ Access to technical expertise and availability of real-time troubleshooting all the time<br>▶ Use of a software with an option for offline data entry |
| | Delays in data entry and missing data | ▶ Training and retraining of researchers in data entry and protocol compliance |
| Intervention management | Difficulty in getting import permits | ▶ Use local commercial or pharmacy suppliers |
| | Drug wastage | ▶ Distribute smaller numbers but frequently replenish study sites |
| | Shipment delays and errors | ▶ Establish an electronic inventory that allows real-time monitoring of drug stock, biospecimen collection and distribution<br>▶ Staff training |
| Adaptive specific | Protocol amendments | ▶ Use an overarching master protocol and consent form and add agent-specific information in appendices to speed up ethical reviews |
| | Handling drug supply for multiple arms | ▶ Use pharmacies that can serve multiple sites within a close geographical area<br>▶ Centralised drug distribution system<br>▶ Choose countries with similar drug labelling requirements |
| | Data entry and analyses | ▶ Predefine stopping rules for each treatment arm<br>▶ Predefine frequency of data analysis |

national coordinating centres.[11 23 53] This arrangement ensured that there were data and regulatory consistencies between countries or research networks.[43 50–52] In general, the CCC consisted of the chief investigator, biostatistician and support staff, being responsible for regulatory oversight, recruitment/retention,[35] trial monitoring,[19 32 44 47 50 51] secondary analyses and communication.[11 23 35 49] Other roles carried out by the CCC included overseeing finance and logistics including drug distribution, storage and analyses of specimen and community support.[11 23 25] Independent overall oversight of was provided by Data Safety Monitoring Board (or Committee; DSMB) and Trial Steering Committee (TSC).[52] Most study reports

referred to the International Conference on Harmonization Good Clinical Practice guidelines (ICH-GCP).[28 34 39 41 42 47 50 51]

## Site and intervention selection

Site selection was often based on initial feasibility assessments.[42 45 48] Several factors were considered during the assessment including local capacity or infrastructure,[7 36 38 42 52] prevalence or burden of the disease in question,[7 38 42] local ethics committees and regulatory processes,[34 39 42 48] individual sites' interest in participating,[7] level of expertise available and insight from local sites representatives.[49] The choice of intervention was usually based on expert reviews[7 43] or a systematic literature review[25 51] which was overseen by the TSC. Some studies also considered treatment guidelines[19 32] and expert consensus where evidence was sparse.[40]

## Recruitment

Recruitment strategies varied across the studies. Factors contributing to the difficulty in enrolling and retaining participants were complex and lengthy screening processes,[27 48] financial constraints,[36] difficulty in maintaining long-term site cooperation[48] and the rigour of navigating varied health systems and setups across countries. Country-specific laws further hampered recruitment: in a German study, participants could only enrol in one trial at a time.[32] Reported strategies that helped to overcome these challenges were frequent communication with participants through a combination of letters, phone calls, mailing of local medical publications containing the trial details and increasing study visibility through the use of websites[33 35 48 53] as well as implementation of the intervention at a convenient location.[33] Some studies organised educational sessions in the form of grand rounds and presentations to healthcare providers which increased visibility.[33 35] Reimbursement of transport costs and keeping visits to a minimum also increased recruitment.[36 43]

## Data management

Data management was not a major challenge and was generally overseen by the Data Coordinating Centre (DCC).[27 29 33 37 39 45 52] Data were usually collected at trial sites using data collection forms and stored locally or in a web-based system. Studies which used the local storage model requested trial sites to securely send data to the DCC where it was amalgamated at regular intervals.[33 37 44] In contrast, use of a web-based system provided real-time data entry, study updates and improved access.[46 50 51 53] Some studies reported technical challenges including problems with internet or firewalls, delays in data collection and missing data.[33 39] This was managed by ensuring sites had access to technical and training support.[27 39]

## Intervention management

### Drug procurement and distribution

Pharmaceutical companies were the most common source of drug procurement,[7 19 26 27 32 37 51 53] and in some cases, the investigational medicinal product (IMP) was supplied free of charge.[19 32 53] In instances where there were drug import restrictions, local pharmacies and suppliers were used.[7 35] Most studies established a clinical coordinating pharmacy to oversee the translation, labelling, repackaging, shipping and coordination of the drug distribution to the study sites.[19 24 25 32 38 47] Country-specific repository centres could be created, particularly in cross-continent studies. These handled the different regulatory and import requirements in addition to providing oversight and support for the distribution of IMPs across sites in their jurisdiction.[19 32 47] More often, studies were faced with difficulty in navigating complex regulatory procedures between countries. For example, some studies in the EU were refused waiver for labelling requirements of a repurposed drug whereas a similar waiver request was granted in the UK.[19 20 28 32] Also, there were country-specific differences in import requirements and issues around 'qualified person' drug release specifications.[19 51] Site pharmacies were involved in drug dispensing in some studies.[26 35 47 48 50 51] SOPs and training were provided to hospital pharmacists at each study site on drug procurement, dispensing, storage, maintenance of accountability logs and disposal of unused or expired drugs.[7 50 51]

## Safety monitoring and adverse events

Serious and adverse events (SAEs) at sites were reported either directly to the CCC or via an electronic data capture or web-based system, then to sponsor who reported to the relevant ethical and regulatory authority. One study ensured that a study coordinator, physician and safety officer were available at all times to manage emergency situations.[35]

All 38 studies established a DSMB for monitoring drug safety and efficacy and safeguarding trial participants. SAEs were reviewed by the independent DSMB at varying intervals based on risk.[7 35 40 43] The timing for these ranged from monthly, to biannually and annually. Some studies used a predetermined interval, for example, in one study the DSMB met every time at least 25–30 participants had a particular data point collected.[25] If there were concerns or SAEs, these meetings were brought forward. All monitoring activities were generally followed by reports which were circulated to all sites.[25 52]

## Biospecimen processing and transport

A variety of biospecimens were often processed in a different country.[27 46 50 53 54] Several studies asked sites to ship the specimen to the designated central facility for analysis.[27 46 50 53 54] Where trials spanned multiple continents, separate facilities were established in each.[24 41 54] Where a central facility for specimen storage was not feasible, a 'virtual biobank' was sometimes established.[52] The collection, handling and processing of specimens were performed in accordance with SOPs.[50 52] Over-collection or under-collection of specimens, increased shipping costs for collection of additional samples, shipments delays, staff shortages and inadequate training were cited

as challenges.[27 49] As most of the specimens were shipped to another country for analysis, regulatory and administrative approvals, coupled with difficulties securing informed consent for international transfer presented additional hurdles.[24 41] Use of an electronic inventory for real-time monitoring of samples, ongoing laboratory training and efficient trial oversight were described as effective mitigation methods.

### Adaptive-specific issues for international trials

Adaptive trial design has become a popular means of increasing the flexibility with which interventions may be interrogated with respect to particular disease indications and/or outcomes, and the efficiency with which this may be achieved (for example through interim analyses in the context of platform trials).[15] They formed a substantial minority of the trials identified in the current review, reflecting challenges raised when implementing them across national boundaries. For example, compared with traditional trials, the need for significantly more documentation to provide clarity on the adaptive process was necessary, including treatment arms, plans for data collection and interim analyses.[25 55] Managing protocol amendments posed a further challenge. Studies reported that using an overarching master protocol and consent form and limiting IMP-specific information to appendices significantly expedited the time for ethics approval when adding or removing treatment arms.[25 52] Handling drug supply for multiple treatment arms while minimising waste was another recurring theme. Studies found it helpful to use pharmacies that could serve multiple clinical sites within a close geographical area, a centralised drug distribution system[25] and choosing countries with similar drug labelling requirements.[55] Data systems that enabled rapid data entry and analyses were another important consideration. Studies reported that setting up a streamlined system with pre-specified stopping rules for each treatment arm was helpful.[34 52] They also had predefined timelines for the frequency of data analysis.[39 52 55]

### DISCUSSION

We conducted a systematic review to determine the operational complexities of conducting international trials, with the aim of providing a useful resource for researchers considering such an approach. Our search strategy employed an extensive array of search terms, themselves falling under four broad 'concepts' and organised to optimise our ability to capture all relevant articles; this approach then necessitated significant refinement during the screening phase and rigorous data collection methods. To this end, each abstract was reviewed by at least two authors for inclusion and 20% of the full-text articles were screened by a second author. Our review highlights that there are various, consistent challenges in the planning, setup, delivery and close out phases of the 38 international trials. Some of the greatest challenges

are posed by ethical and regulatory obstacles, a lack of harmonisation within EU and between EU and other developed countries being a key element. Similar findings have been highlighted in another systematic review on conducting trials in developing countries.[56] Streamlining funding and regulatory processes is one possible solution. The EU Clinical Trials Regulation launched a portal in 2022 that enables registration of trials with sites in up to 30 EU countries in a single platform. This system also resolves data sharing issues, providing a one-off consent and enabling national regulators to collaboratively process approvals.[57] However, there is no obligation for EU countries to participate until 2025 and this process does not provide a solution for collaboration with other continents. Furthermore, this portal does not extend to UK sites following its withdrawal from the EU. Possible interim solutions to global collaboration include defining responsibilities of sponsors and legal representatives, ascertaining insurance requirements prior to country selection and ensuring all sites are provided with a framework for the regulatory review process. In addition, working with a specialist insurance broker and careful country selection based on previous successful collaborations have also been suggested.[58] Establishing a national support group to provide mentoring to less experienced trialists for the entire trial process may also aid in the process.[56] By contrast, previous literature has described other key challenges including lack of infrastructure, poor data quality and lack of training particularly in resource-limited settings. These were not a particular challenge in our systematic review.[56 59]

Budgetary constraints and contract negotiations were described as further hurdles. Over the last few years, several networks have been formed to increase funding opportunities. For example, INSIGHT was created to provide governance and funding support for multinational HIV treatment trials but has expanded to also include influenza and COVID-19.[60] NeuroNEXT (Network for Excellence in Neuroscience Clinical Trials) promotes partnership in neurological disease trials by providing a centralised system for ethics review, contracting agreements and data management but is limited to multicentre studies in the USA.[61] Use of these networks, establishing site set-up costs during the trial planning phase and timely contract negotiations may assist in this process.

Managing drug distribution across international borders remains a challenge, as has also been previously reported.[21] Tailoring supply chains while minimising drug wastage is clearly desirable. Establishment of central drug repositories in each continent and using pharmacies which can serve multiple sites rather than the traditional one-pharmacy-one-site model has been shown to be effective.[19 47] Involving pharmaceutical partners in drug distribution is another possibility which also eliminates the need for time-consuming vendor selection.[62]

Our search strategy was employed an extensive array of search terms, themselves falling under four broad 'concepts' and organised to optimise our ability to capture

all relevant articles; this approach then necessitated significant refinement during the screening phase to arrive at robust and our data collection methods rigorous. Each abstract was reviewed by at least two authors for inclusion and 20% of the full-text articles were screened by a second author.

## Limitations and future work

The extent to which international facets of clinical trials are made explicit in their description, and the terminology used in the literature to describe clinical trials in general, have evolved over time, and this observation is reflected in the pragmatic search and screening strategy we adopted. As with any systematic review, we cannot exclude the possibility that decisions about individual terms included in, or excluded from, our searches may have influenced the precise range of papers pertaining to individual international trials that would have been captured in our systematic review—and hence, the challenges and solutions identified during data extraction. Linked to this, not all investigators routinely publish information on challenges encountered in trial design and conduct, meaning our findings cannot be said to be exhaustive. Moreover, our systematic review only focused on trials conducted in more than one country and studies published after 2005 and therefore our results may not be generalisable to multicentre studies conducted within the same country. For example, ethics regulations, sponsor responsibilities and insurance requirements are less likely to be a hurdle in single nation trials.[62] With respect to potential solutions to the challenges described, these were extracted from included papers where identified in association with specific challenges; while every effort was made to mandate this link, an element of subjectivity when summarising such solutions in the current report cannot be excluded. Finally, we did not assess clinical outcomes of included studies, considering this beyond the scope of our endeavour. Further research in the form of a qualitative or Delphi study with trialists and key stakeholders may provide more in-depth information on these challenges and possible actions to mitigate them. We also urge researchers involved in conducting international trials to routinely report operational challenges.

## CONCLUSION

International trials address many unmet needs in trial design but their proponents still face operational challenges at every level, ranging from difficulties with funding and obtaining regulatory approvals to site contracts and drug distribution. Careful planning and communication with sites and key stakeholders during the trial planning phase can overcome delays presented by some of these challenges. More generally, given the upsurge in global trials particularly since the COVID-19 pandemic, recognition by policymakers of the potential rewards that regulatory harmonisation between nations could bring for the delivery of more efficient and cost-effective research should positively impact the health and well-being of their citizens. National and international organisations should continue to work collaboratively to develop infrastructures that support international trials.

**Author affiliations**
[1]Translational and Clinical Research Institute, Newcastle University, Newcastle upon Tyne, UK
[2]Northumbria Healthcare NHS Foundation Trust, Northumbria, UK
[3]NIHR Innovation Observatory, Population Health Sciences Institute, Newcastle University, Newcastle upon Tyne, UK
[4]Population Health Sciences Institute, Newcastle University, Newcastle upon Tyne, UK
[5]Newcastle Clinical Trials Unit, Newcastle University, Newcastle upon Tyne, UK
[6]Musculoskeletal Unit, Newcastle Upon Tyne Hospitals NHS Trust, Newcastle Upon Tyne, UK

**Acknowledgements** We would like to thank Chris Price, Andrew Johnston, Julia Philipson and Ruth Wood for providing insights into outcome measures.

**Contributors** LG was involved in conception of the study, screening, data extraction, analysis and writing up the manuscript. OA was involved in screening and writing up the manuscript. AI was involved in developing search strategy and revising it critically for important intellectual content. RF and MS were involved in screening and revising it critically for important intellectual content. MB was involved in developing search strategy and revising it critically for important intellectual content. LO, JL-K, JDI, JW and DC co-conceived the study and revised the manuscript for important intellectual content. AGP co-conceived the study and revised the manuscript for important intellectual content; he accepts full responsibility for the work, had access to the data and sanctioned the decision to publish.

**Funding** This study was funded by National Institute of Health and Care Research (NIHR153955). Infrastructural support was provided by the National Institute of Health and Care Research (NIHR) Newcastle Biomedical Research Centre, and to JDI and AGP via the Research into Inflammatory Arthritis Centre Versus Arthritis (RACE; grant number 22072). JMSW is funded by an NIHR Research Professorship (NIHR301614). LG is funded by a NIHR Academic Clinical Fellowship. The views expressed are those of the author(s) and not necessarily those of Versus Arthritis, the NIHR or the Department of Health and Social Care.

**Competing interests** None declared.

**Patient and public involvement** Patients and/or the public were not involved in the design, or conduct, or reporting, or dissemination plans of this research.

**Patient consent for publication** Not applicable.

**Ethics approval** Not applicable.

**Provenance and peer review** Not commissioned; externally peer reviewed.

**Data availability statement** No data are available. No additional data available.

**ORCID iDs**
Leher Gumber http://orcid.org/0000-0003-0154-1207
James MS Wason http://orcid.org/0000-0002-4691-126X
Arthur G Pratt http://orcid.org/0000-0002-9909-8209

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
