## [Reviewer comments · BMJ Open]

ARTICLE DETAILS

TITLE (PROVISIONAL)	Operational complexities in international clinical trials: a systematic review of challenges and proposed solutions
AUTHORS	Gumber, Leher; Agbeleje, Opeyemi; Inskip, Alex; Fairbairn, Ross; Still, Madeleine; Ouma, Luke; Lozano-Kuehne, Jingky; Bardgett, Michelle; Isaacs, John; Wason, James; Craig, Dawn; Pratt, Arthur

VERSION 1 – REVIEW

REVIEWER	Julia M. Machline-Carrion ePHealth Primary Care Solutions
REVIEW RETURNED	07-Jul-2023

GENERAL COMMENTS	The manuscript entitled “Operational complexities in international clinical trials: a systematic review of challenges and proposed solutions” presents a systematic review aiming to identify the operational complexities of conducting international clinical trials and to identify potential solutions for overcoming them. In times of urgent need for efficient and informative evidence generation for clinical practice, understanding pitfalls in trial management and identifying strategies to overcome barriers is key. However, some aspects deserve considerations: 1) Inclusion criteria and Study Selection: a. While the abstract mentions “any publication reporting operational challenges”, the methods section mentions that the search “covered four broad areas: (i) international trials; (ii) adaptive design trials; (iii) study design; (iv) specific challenges and requirements of designing and running trials”, thus it is not clear what types of manuscripts were targeted (just trial original research? review articles?). b. Did the authors consider retrieving information from trial registry databases (e.g. ClinicalTrials.gov)? c. When reviewing the appendix, one has the impression that the search was targeting adaptive trials over other designs. This could explain the relatively low number of studies reporting operational constraints... Could the authors please clarify this point? 2) Methods: Which operational challenges were considered? Was there any definition used? This should be better described. a. The predefined proforma used for data extraction should be included in the supplementary material. b. Since one of the objectives was to identify potential solutions for overcoming operational complexities, how was this information extracted? Was this reasoning developed solely on authors' argumentation or was this information extracted from the studies? This should be clarified. c. How was the funding source classified? Since multiple sources may coexist, how was the main one identified? 3) Results: Table 1 is a very important data source. It would be desirable to add other informative data that might be related to
--

	operational constraints such as : the leading country (the country that hosts the coordinating center) , number of different regulatory agencies involved, presence of a contract research organization, sex, and gender of the leading investigator). 4) Discussion: Considering the complexity of the subject, the discussion is too brief and superficial. Of Importance, this section focuses mainly on the proposed solutions (which are not clear what source was used to identify them) than on then study findings. Additionally, no study limitations were addressed. It is desirable that the authors address limitations, as a matter of transparency and also to show the readers the extent of reasoning the investigators have reached on the subject.
--	--

REVIEWER	Luca Sforzini King's College London
REVIEW RETURNED	23-Jul-2023

GENERAL COMMENTS	This is a very interesting paper addressing an important research topic. A systematic appraisal of operational complexities in international clinical trials may provide important insights into the way clinical research is conducted, and assist in the development of future studies. I commend the authors' efforts in producing this manuscript. I recommend minor revisions before endorsing the publication of the paper. In the introduction, the authors provide broad information about international clinical trials. They rightfully include platform trials in their review. I believe further discussion of what platform trials are and how these differ from standard RCTs would be adequate. They also identify some challenges for studies conducted under a master protocol. This is further elaborated in a section on adaptive-specific issues. However, there are still significant clinical advantages to a platform approach, and in having a single overarching protocol, rather than completely different trials. These considerations could be included in a more balanced discussion. In addition, I would consider including trial designs (such as platform) in table 1 for a better clarity. I would suggest that the authors put more emphasis in their discussion on the proposed solutions (something they also mention in the title). Also, recommendations for future research would be useful. Lastly, even though I understand that it may have not been within the aim of the research, the paper does not consider the clinical outcomes of the trials. This is clearly something that should be considered and balanced with operational complexities when designing clinical studies, and I suggest the authors acknowledge this as a limitation.
---

REVIEWER	Matthew Sydes Medical Research Council Clinical Trials Unit at University College London, MRC Clinical Trials Unit, Institute of Clinical Trials and Methodology
REVIEW RETURNED	16-Nov-2023

GENERAL COMMENTS	The aim of the review was not at all clear to me, I'm afraid. The title
---

	is about “international clinical trials” as are the first lines of the introduction and discussion. The very first line of the Introduction (without a reference) implies that people have only recently started to do international trials whereas international trials have been going on for decades. There have been thousands of international trials. Why are there only 38 in the review?. Perhaps it’s because the criteria are actually narrower, but I didn’t feel the Introduction really sets out why this paper would be needed nor quite what its scope would be. The search criteria set out in the manuscript don’t give me a good feel for what sort of papers were hoped for: was it trial results, trial protocols or papers dedicating to reflecting on operational experiences. If it’s “just” international, I’m aware of at least one dedicated paper reflecting on operationalising an multi-national trial that was missed or was unexpectedly out of scope. If it’s international and adaptive, there are papers from 2019 I might have expected to be included given other papers that have been selected. Some of the terminology around adaptive and platform protocols has shifted over times: many international trials that addressed multiple questions perhaps would not have been identified as platform protocols in the past (even if they had early stopping rules) but probably would be now. The search criteria are extensive but difficult to read. Some form of annotation and chunking might help more readers understand quite what’s going on. Are the experiences people choose to write about (if that’s what this review is doing) somehow different to those that people don’t write about? If experiences from one trial have appeared in multiple papers, are they over-represented here or might lessons have been missed because only one manuscript was permitted in. One paper I was involved in has made the cut, but it’s not the only paper that reflects on implementation, and another paper from the same trial that reflects specifically on international challenges has not been selected. Without being able to be sure I know what the researchers were looking for (and why), it’s difficult to know the extent to which they are succeeded (I might be wrong as to whether the papers I expected to see should be there - it's not clear enough to me that I should know); so I cannot be sure how to interpret the Results or Discussions at this stage (even though it looks like that section contains some useful information.)
--	---

VERSION 1 – AUTHOR RESPONSE

Reviewer 1		
Comment raised	Response	Location of revisions
Inclusion criteria and Study Selection: a. While the abstract mentions “any publication reporting operational challenges”, the methods section mentions that the search “ covered four broad areas: (i) international trials; (ii) adaptive design trials; (iii) study design; (iv) specific challenges and requirements of designing and running trials , thus it	As now made clearer, when adopting the broad database search strategy and subsequent screening approach outlined, studies describing operational challenges related to international trials were sought. Studies excluded at screening included those (whether or not adaptive trials) conducted in only one country, those in which operational challenges were not	Revised “Search strategy and study selection” section (Methods), Figure 1A and Supplement.

is not clear what types of manuscripts were targeted (juts trial original research? review articles ?).	described, systematic reviews, abstracts, protocols, and studies not available in English.	
b. Did the authors consider retrieving information from trial registry databases (e.g. ClinicalTrials.gov)?	We did not use ClinicalTrials.gov during our database search. We indeed considered this but the outcomes of interest were often qualitative in nature and rarely discernible on review of ClinicalTrials.gov or other trial registries.	N/A
c. When reviewing the appendix, one have the impression that the search was targeting adaptative trials over other design. This could explain the relatively low number of studies reporting operational constraints ... Could the authors please clarify this point?	This is not the case and we agree that clarification was needed; we hope the above clarifications (including the conceptual overview now depicted in Figure 1A) satisfy the reviewer: the emphasis is on international trials.	Revised introduction, revised "Search strategy and study selection" section (Methods), Figure 1A and Supplement.
Methods: Which were the operational challenges considered ? Was there any definition used ? This should be better described.	These were broadly defined as any challenges in the set up and conduct of international trials. We extracted data on the multiple operational challenges in relation to specific elements now listed more explicitly in Supplementary Material ("[a]; specific challenges") and referred to in the "Search strategy and study selection" section (Methods).	"Search strategy and study selection" section (Methods) and Supplementary Material.
a. The predefined proforma used for data extraction should be included in the supplementary material.	We have added this information to the supplementary material, now also referred to in the "Data extraction" section.	"Data extraction" section (Methods) and Table S1
b. Since one of the objectives was to identify potential solutions for overcoming operational complexities, how were this information extracted ? Was these reasoning developed solely on authors argumentation or was this information extracted from the studies? This should be clarified.	Whilst we accept that an element of subjectivity is impossible to exclude in the process – a factor we now acknowledge as a limitation in our abstract/discussion – solutions to identified challenges were extracted directly from the included studies where they were described, and every effort was made to limit subjective interpretation in this context.	Abstract and Discussion (limitations).

c. How was the funding source classified ? Since multiple sources may coexist, how the main one was identified ?	Information on funding sources was obtained directly from the description(s) provided in the included studies. It was sub-categorised into government, academic (e.g. university), charity and industry. Where doubt existed and/or more than one funding body was mentioned, consensus between two reviewers was reached as to the predominant funding source. We have amended the text to clarify this (“Definition of variables and data analysis” section).	“Definition of variables and data analysis” section (Methods).
3) Results : Table 1 is a very important data source. It would be desirable to add other informative data that might be related to operational constraints such as : the leading country (the country that hosts the coordinating center) , number of different regulatory agencies involved, presence of a contract research organization, sex, and gender of the leading investigator).	We agree that information on the location of coordinating centre (which was captured during our data extraction) is relevant and have added this as an additional column to Table 1 as requested. Whilst the sex (but not gender) of the lead author of studies could be inferred from authorship details of published studies in most cases – and whilst we can see that such information may be relevant in specific circumstances – we do not feel it is relevant for purposes of the current systematic review given its scope. Other requested elements were unfortunately not available/extracted.	Table 1
4) Discussion: Considering the complexity of the subject, the discussion is too brief and superficial. Of Importance, this section focuses mainly on the proposed solutions (which are not clear what source was used to identify them) than on then study findings. Additionally, no study limitations were addressed. It is desirable that the authors address limitations, as a matter of transparency and also to show the readers the extent of reasoning the investigators have reached on the subject.	We agree that a more detailed section on study limitations was warranted, and we have added this as requested. We furthermore draw the reviewer’s attention to an additional line of explanatory text in the “data extraction” section (Methods).	“Data analysis” section (Methods) and Discussion

Reviewer 2		
Comment raised	Response	Location of revisions
In the introduction, the authors provide broad information about international clinical trials. They rightfully include platform trials in their review. I believe further discussion of what platform trials are and how these differ from standard RCTs would be adequate. They also identify some challenges for studies conducted under a master protocol. This is further elaborated in a section on adaptive-specific issues. However, there are still significant clinical advantages to a platform approach, and in having a single overarching protocol, rather than completely different trials. These considerations could be included in a more balanced discussion. In addition, I would consider including trial designs (such as platform) in table 1 for a better clarity.	In the modified opening paragraph of the introduction we now include a working definition of platform trials. We agree that some additional discussion of “pros” (as well as complexities) of such studies is appropriate and have added some lines to the specified section. We have furthermore added a column on trial design to Table 1.	Introduction (first paragraph); Results (section on “adaptive-specific issues”; Table 1
I would suggest that the authors put more emphasis in their discussion on the proposed solutions (something they also mention in the title). Also, recommendations for future research would be useful.	Additional lines have been inserted to address this comment as suggested.	Discussion
Lastly, even though I understand that it may have not been within the aim of the research, the paper does not consider the clinical outcomes of the trials. This is clearly something that should be considered and balanced with operational complexities when designing clinical studies, and I suggest the authors acknowledge this as a limitation.	The reviewer is correct that the clinical outcomes of the trials described was not within its scope and, given the diversity of target populations and objectives represented by the international trials included in our systematic review, it would, we feel, be difficult to provide a meaningful interpretation of these without distracting from the	Limitations section of discussion

	main focus of the manuscript. Nonetheless, we now acknowledge this issue in our discussion.	
Reviewer 3		
Comment raised	Response	Location of revisions
The aim of the review was not at all clear to me, I'm afraid. The title is about "international clinical trials" as are the first lines of the introduction and discussion. The very first line of the Introduction (without a reference) implies that people have only recently started to do international trials whereas international trials have been going on for decades. There have been thousands of international trials. Why are there only 38 in the review?. Perhaps it's because the criteria are actually narrower, but I didn't feel the Introduction really sets out why this paper would be needed nor quite what its scope would be. The search criteria set out in the manuscript don't give me a good feel for what sort of papers were hoped for: was it trial results, trial protocols or papers dedicating to reflecting on operational experiences. If it's "just" international, I'm aware of at least one dedicated paper reflecting on operationalising an multi-national trial that was missed or was unexpectedly out of scope. If it's international and adaptive, there are papers from 2019 I might have expected to be included given other papers that have been selected.	We have modified the introduction based on this critique, removing any unintended suggestion international trials were novel per se, and amending/adding to the final sentence describing our aim – to identify operational complexities (and their potential solutions) in respect of the conduct of international trials. This, we feel, goes some way to accounting for the smaller number of papers reviewed than anticipated by the reviewer. We furthermore hope that our general response to our reviewers, above, together with the amended text and clarifications outlined, sufficiently clarify our search and selection approach for the general reader. It is difficult to comment on specific but unidentified "omissions" the reviewer has concerns about, but reference to the revised elements should help to do so.	Revised introduction, revised "Search strategy and study selection" and "data extraction" sections (Methods), Figure 1A and Supplementary Material.
Some of the terminology around adaptive and platform protocols has shifted over times: many international trials that addressed multiple questions perhaps would not have been identified as platform protocols in the past (even if they had early stopping rules) but probably would be now. The search criteria are extensive but difficult to read. Some form of annotation and chunking might help more readers understand quite what's going on.	The reviewer identifies an important challenge we faced when designing our search/screening strategy. Indeed, recognising evolving terminologies, the broad range of terms employed by our search strategy was adopted precisely to capture as many relevant articles as possible, but necessitated significant refinement during	revised "Search strategy and study selection" and "data extraction" sections (Methods).

	screening, as outlined.	
Are the experiences people choose to write about (if that's what this review is doing) somehow different to those that people don't write about? If experiences from one trial have appeared in multiple papers, are they over-represented here or might lessons have been missed because only one manuscript was permitted in. One paper I was involved in has made the cut, but it's not the only paper that reflects on implementation, and another paper from the same trial that reflects specifically on international challenges has not been selected. Without being able to be sure I know what the researchers were looking for (and why), it's difficult to know the extent to which they are succeeded (I might be wrong as to whether the papers I expected to see should be there - it's not clear enough to me that I should know); so I cannot be sure how to interpret the Results or Discussions at this stage (even though it looks like that section contains some useful information.)	Again, we are unable to comment upon concerns about specific but unidentified publications the reviewer was involved in, but his general point is pertinent and well-taken. We feel it can best be addressed by explaining and rendering our search/screening strategy as transparently as possible; we acknowledge deficiencies highlighted by the reviewer in this regard and hope the concern has been adequately addressed in our manuscript's revised form. Inevitably, decisions about individual terms included in, or excluded from, our searches may have influenced the precise range of papers pertaining to individual international trials that would have been captured in our systematic review. We acknowledge the potential limitation in the discussion of our revised manuscript. Given its over-arching aim – to deliver a reference of practical value to prospective trialists planning to set up international trials in the modern era – we consider this impact on our report's findings is likely to be modest.	General responses, abstract and discussion (limitations).

VERSION 2 – REVIEW

REVIEWER	Julia M. Machline-Carrion ePHealth Primary Care Solutions
REVIEW RETURNED	14-Feb-2024

GENERAL COMMENTS	Although the authors have attempted to answer the comments, there are still major doubts concerning the methods and definitions applied.
--

REVIEWER	Luca Sforzini King's College London
REVIEW RETURNED	30-Jan-2024

GENERAL COMMENTS	The authors addressed all my concerns. I believe the paper clearly states strengths and limitations, with some important findings for future research. I have no additional comments.
---